# The Numerical Modelling Approach with a Random Distribution of Mechanical Properties for a Mismatched Weld

**DOI:** 10.3390/ma14195896

**Published:** 2021-10-08

**Authors:** Luka Starčevič, Nenad Gubeljak, Jožef Predan

**Affiliations:** Faculty of Mechanical Engineering, University of Maribor, Smetanova 17, SI-2000 Maribor, Slovenia; luka.starcevic@student.um.si (L.S.); jozef.predan@um.si (J.P.)

**Keywords:** weld metals, welded joints, damage mechanics, finite element analysis, crack growth, ductile fracture

## Abstract

The aim of this work was to include a local variation in material properties to simulate the fracture behaviour in a multi-pass mis-matched X-weld joint. The base material was welded with an over and under-match strength material. The local variation was represented in a finite element model with five material groups in the weld and three layers in the heat-affected zone. The groups were assigned randomly to the elements within a region. A three-point single edge notch bending (SENB) fracture mechanics specimen was analysed for two different configurations where either the initial crack is in the over or under-matched material side to simulate experimentally obtained results. The used modelling approach shows comparable crack propagation and stiffness behaviour, as well as the expected, scatter and instabilities of measured fracture behaviour in inhomogeneous welds.

## 1. Introduction

Many researchers [1,2,3,4,5,6] who deal with numerical simulations of the strength and fracture behaviour of welds are looking for a suitable numerical universal tool to describe as faithfully as possible the behaviour of a weld with a crack. In particular, they focus on crack propagation through different strength weld materials, as, in the case of confirmation of the correctness of these tools, simulations can be performed for different weld shapes and for different materials and different loading methods [7,8,9].

Welded joints represent heavily inhomogeneous material regions of structures, which result in a local crack driving force and in a crack path deviation, where a crack propagates through different strength regions. The effect is also reflected globally in the force vs. displacement load curve. Many researchers [10,11,12,13,14,15,16] have investigated the influence of the material properties’ inhomogeneities in a welded joint on fracture behaviour using experimental and numerical methods. They developed an approach for local crack driving force determination based on the configurational force concept [17]. The local crack driving force is calculated by post-processing followed by a classical finite element analysis as the sum of a far-field crack driving force and additional material inhomogeneity term. Many studies have been published for different material inhomogeneity configurations and spatial variations in material properties. They studied the effect of material inhomogeneities for discrete jumps of material properties at the interfaces, as well as continuous variation in properties in biomaterials. In the numerical simulations, they were focused on the point of crack initiation of the stable crack growth [16], where they obtained a good match between the experimental and numerical results, but, in the case of crack growth, they received significant deviations due to the crack deviation from the initial pre-fatigue crack plane.

Globally, distinct strength inhomogeneous welds are repair multi-pass welds in high loaded structures, where the part of the weld with the defect must be removed by grooving and filling with an under-strength filler material. If hidden defects such as pores or non-melted situ occur during repair welding, a crack is initiated in the low-strength weld material and propagates towards the high-strength part of the weld. Globally, two regions with different material properties affect the local crack driving force magnitude and direction, which influence crack growth rate and deviates the crack path from the initial straight. Some fracture instabilities can be caused by the extremely increased local crack driving force by the material inhomogeneity in the cases where the crack propagates from the over to under-strength material, and vice versa, the crack can be arrested by the diminishing local crack driving force in the case where the crack approaches the interface from the lower strength material.

To ensure the structure integrity of the weld in the presence of a crack, it is important to estimate the residual load capacity through the force displacement load curve.

The purpose of the study is to present the numerical simulation results of the load vs. crack mouth opening displacement (CMOD) curve for the propagating cracks through globally and locally inhomogeneous welds, taking into account the local mechanical properties obtained from the standard and mini tensile specimens (MTSs), as well as with the empirical correlation between the microhardness and strength.

The subject of the numerical simulation is the fracture behaviour of a multi-pass inhomogeneous weld consisting of two different filler materials with an initial crack in the under-strength weld part growing towards the over-strength half and with an initial crack in the over-strength weld part growing towards the under-strength weld half. 

It is well-known that weld joints have inhomogeneous mechanical properties. These usually appear in multi pass mismatched welds, where the properties are combined in order to achieve the desired fracture behaviour. Typically, the combination of mechanical properties in a mismatched weld are the following: one half under (UM) and one half over (OM) matched weld material, and on both sides of the heat affected zones (HAZs) and base material (BM). Usually, the weld material should be an OM weld metal in order keep the OM material elastic, while plasticity starts in the BM. Therefore, the higher probability to failure is expected in the BM or HAZ than the OM weld metal. The combination of selected weld materials can, therefore, affect the stiffness response and crack paths of the weld joint significantly. Thus, the structural integrity of the cracked mismatched weld joint depends mainly on the fracture toughness of the cracked zone and loading condition [18,19,20].

The local mechanical properties at the BM, HAZ and inhomogeneous weld should be considered in order to consider the structural integrity for designing the weld structures. In the past, detailed experimental investigations were carried out for multi pass welds to analyse the local mechanical performance inside the weld and HAZ [21,22,23,24,25] in the SENB specimen. It is known that mechanical properties such as spatial yield stresses vary within the hardener and softer zones (HAZs) and the weld region. Therefore, this local material inhomogeneity should be considered in the finite element (FE) simulation, as in the latest approaches [26,27,28]. 

However, a sufficient approach for modelling a multi pass weld does not exist yet, and is required to analyse failure potentials (crack paths) for welded structures. With such a model, critical welds inside large structures (pressure vessels, welded components, etc.) can be analysed and optimised to reach damage tolerant behaviour.

## 2. Materials and Experiments

In our case, we focused on two materials deposited in multi pass “X” -welded joints, with two crack configurations, either the initial crack in the UM (configuration 1) or OM (configuration 2) weld site, according to Figure 1 and Table 1. NIOMOL 490 was used as a base metal (BM), FILTUB 75 as an OM and VAC 60 as UM materials for the weld.

The material NIOMOL 490 is a high-strength low-alloy fine grain steel, with retard to coarse grain growing in the heat affected zone. Therefore, NIOMOL 490K has very good weldability and it is possible to welded without preheating with low strength consumable materials, e.g., VAC60. The mechanical properties and chemical compositions of the BM, the OM and UM weld metals are provided in Table 2 and Table 3. A flux cord arc welding (FCAW) procedure was applied, and two different tubular wires were selected for welding in order to produce welded joints in over- and under-matched (OM and UM) configurations. The heat input of each weld pass was between 15 and 18 kJ/cm, corresponding to the cooling time between 500 and 800 °C ∆t_8/5_ = 8–12 s. Such weld metal configurations are common for repairing welding.

To determine the tensile behaviour, experimental testing of five specimens was carried out to collect the yield strengths R_p02_ and R_m_ as a reference for the material model development.

The base material properties were kept constant, while the weld metal properties varied. This variation is described by the mismatch factor:(1)$$M=\frac{\sigma_{\mathrm{YW}}}{\sigma_{\mathrm{YB}}}$$
where $\sigma_{\mathrm{YW}}$ and $\sigma_{\mathrm{YB}}$ present the yield strength of the weld metal and the yield strength of the base metal, respectively. The weld metal is commonly produced with a yield strength greater than that of the base plate; this case is designated as overmatching (OM) with the mismatch factor *M* > 1. However, an increasing use of high-strength steels forces the fabricator to select a consumable with lower strength to comply with the toughness requirements, which are designated as under-matching (UM), where *M* < 1.

Later, the local variation in the tensile behaviour was tested and analysed in the weld for the UM and OM weld material, as well as the HAZ, by using a set of mini tensile specimens (MTSs). The orientation and position of both set of specimens is shown in Figure 2. MTSs are fabricated by wire spark eroding techniques. MTS testing was performed by uniaxial testing under a constant stroke velocity of 0.1 mm/min and by laser extension measurement, with an initial length of 8 mm.

The local mechanical properties for the OM and UM weld material and corresponding HAZ are presented in Figure 3.

With the presented sample, a three-point SENB specimen was analysed in accordance with the standard ASTM E1820. The sample thickness was W = 25 mm, and the initial crack length a_0_ = 11.2 mm (configuration 1) or 7.9 mm (configuration 2). Figure 4 shows schematic view of specimen orientation in a welded plate and testing manner. The fracture toughness testing was performed at room temperature, 24.5 °C, and under a constant stroke velocity of 0.5 mm/min. The CMOD versus the reaction force F was recorded and compared with the simulation results. The experiments are performed at room temperature (+24 °C) for standard, mini tensile and three point bending specimens by the servo-hydraulic testing machine INSTRON. A single specimen method was used for the crack tip opening displacement (CTOD) testing according to the standard BS 7448 [29]. The CTOD tests were performed under displacement control at a loading rate of 1 mm/min. The load (F), the load point displacement, and the crack mouth opening displacement (CMOD) were recorded. Figure 10a shows the experimentally obtained F vs. CMOD load curve where, after a certain amount of stable crack propagation in the OM metal and after achieving the maximum load, a step of unstable crack propagation is exhibited, followed by stable crack growth in the UM metal.

## 3. Finite Element Simulation

A two-dimensional model of a three-point SENB specimen was modelled according to the testing procedure and specimen geometry, as is shown in Figure 5. All simulations were performed using a commercial finite element method software, SIMULIA Abaqus [30], an implicit dynamic solver with the quasi-static application (quasi-static loading also applied in hardware). The FE model was assembled with the specimen-welded structure and the loading roller (16 mm in diameter) as an analytic rigid body. Displacement over time was defined on the upper roller. The supports on both sides were modelled with the prescribing boundary conditions (y = 0) at two nodes; therefore, neighbouring elements had local linear elastic material definition due to stress concentration issues. Therefore, the FE model complexity and computation time were reduced, but the results were not affected by the simplification.

The analysed samples are discretised with plane strain finite elements (first order, CPE4 in the region of interest and CPE3 in the not interested regions). A structured mesh with quad elements (size 0.25 mm) was applied between the weld materials and the HAZ. Towards the outside, the element size increased up to 2.0 mm in length, since in this region, the influence on the overall behaviour can be neglected (Figure 6). The FE model consisted of 18,364 finite elements with 18,525 nodes and 18,359 finite elements with 18,525 nodes, as is referred in Table 4, for initial crack in OM and initial crack in UM, respectively. The model thickness was B = 25 mm as it was on the tested sample.

The analysed model for configuration 1 consisted of base material, and over- and under-matched material (Figure 5). The initial notch was one finite element wide; the notch length was according to the fatigue pre-crack length in Section 2. Since the mechanical properties within the HAZ change with the distance from the weld interface, three layers of equal thickness of HAZ were defined to describe the material properties’ variation in the HAZ. Table 5 shows average material properties. 

In addition to the elastic-plastic material model, the ductile damage formation [31] was considered for the BM, OM and UM materials. All material parameters (Young’s module, Poisson’s ratio, plastic strain hardening curve, plastic strain at damage initiation and critical plastic displacement) were defined according to the uniaxial tensile testing response in Section 2. We used an elastic-plastic ductile damage material model to describe material behaviour because the comparison between curves from tensile experimental testing (full lines) and simulations (dotted lines) showed excellent agreement with each other, and they are plotted in Figure 7. The material damage properties were calibrated for the finite elements size of the tensile specimen model, and the same size was used for the three-point SENB specimen.

The used material models were the base for the material models defined inside the weld and HAZ. As presented above, the global mechanical properties were inhomogeneous in the model, and exhibited a slight local variation inside each material region. Our main goal was to use a modelling technique that includes small local variations in the measured values inside global material regions. The curve shapes were used for given regions and scaled according to the variation in the yield strength Rp02 measured with micro tensile specimens.

We focused on the R_p02_ and R_m_ values and created five groups for the UM and OM material with respect to how many measured points were inside each group-shares on each level, as seen in Table 6. Figure 8 illustrates the grouping based on the R_p02_ values. Therefore, five material models were created for the UM and OM, and they were based on the reference curve model, as shown in Figure 7, where the true plastic-stress values and damage parameters were scaled according to Table 6. The length interval of each group was proportional to the frequency of properties on the strength level. Further, all elements inside a weld material were assigned randomly to a property, with respect to the shares of each material/group. Figure 9 show the random distribution of elements, with five different mechanical properties for both specimens with different two-filled material properties.

The HAZ was modelled with three equal thick layers (1.25 mm wide, Figure 9) according to the variation in material properties in the HAZ, see Table 5 and Figure 3, with scaling of the plastic-stress values and damage parameters. The input parameters for the scaling were the averages of the R_p02_ and R_m_ values from both sides of the weld. An example of the FE-Model with five groups for the UM/OM weld and additional three layers for the HAZ is presented in the Figure 9.

## 4. Results

The reaction force on the middle loading roller and the CMOD were measured and compared between the experimental and simulation responses for the three-point SENB specimen. As mentioned above, the material properties were assigned randomly to the elements inside each of the two weld regions; therefore, three models were analysed to show the influence of the random distribution/assignment.

Figure 10 shows the reaction force versus CMOD for the two configurations compared with the experimental curves. The behaviour between FE simulation and experimental testing showed a similar response along the whole loading and unloading sequences, as well as some instabilities came out from the simulation, as they appeared during fracture mechanics testing.

Figure 10a shows the case when the crack start propagates from the UM along the symmetry line and then bends on the interface between the UM and OM material region and continues along the interface between OM and the neighbouring HAZ region for the UM/OM configuration. Nearly the same observations were seen in the fracture mechanics experiment. The three numerical models (configuration UM/OM) with different material properties distribution, predicted different crack propagation paths due to the slightly different material properties distribution. Nevertheless, the stiffness behaviour predicted by the three random distributions was very similar.

In order to present the stress state at characteristic points of loading F vs. CMOD, for each numerically obtained curve five characteristics points were selected, marked by I.–V. Figure 11a I. Shows that the highest von Mises stress concentration appeared far from the crack tip and behind the fusion line between the UM and OM weld metal. Figure 11a II. shows that the crack path follows the maximum von Mises stress path (left or right) from the crack tip to the HAZ-OM-UM triple point, where the crack turned and propagated between the over-matched weld metal and the HAZ. Figure 11a from III. to VI. shows that the maximum von Mises stress remained at the crack tip between the OM and HAZ fusion line. Figure 11b I. shows that the maximum von Mises stress appeared at the crack tip in the OM weld metal. At the point of maximum sustained loading (Figure 11b) II. a maximum von Mises area appeared in the OM and stable crack growth straight to the fusion line between the OM-UM weld metal, as shown in Figure 11b III.–VI. The numerically obtained results, which were compared with the experimental fracture behaviour of the three-point SENB standard specimens by ASTM E-1820, showed a good agreement on the load vs. crack mouth opening displacement curves, as well as a comparable match between the numerically simulated and metallographic measurement deviation of the crack paths. The simulation results show that the fracture behaviour of a three-point SENB specimen with a crack in the middle of a globally heterogeneous weld with good agreement with the experimental results can be described on the basis of tensile stress-strain curves obtained from standard, and scaled with mini-tensile specimens, for each material microstructure.

Figure 12 shows crack paths for both specimens made by polishing and etching of both halves of the tested specimens.

## 5. Conclusions

A numerical investigation was carried out to analyse the effect of local variation in material properties to simulate fracture behaviour in a mismatched X-weld joint. We simulated the fracture behaviour of the strength mismatched multi pass welds numerically successfully by using global and local variations in material properties. The input data were the mechanical material property curves measured from the standard and mini-tensile tests. In the simulations we used the elastic-plastic and ductile damage model in the ABAQUS [31] software by arranging five local areas randomly for varying local properties. The random variation in local material properties where the proportion of one property remains constant did not cause significant deviations between the results of the numerical simulations up to the maximum load for both simulated configurations. The response curves differed after maximum load during the damage process, and the same deviation appeared from the experimentally obtained response. We demonstrated that the response mainly has an effect on the local variation in the properties, which evidently appeared in the multi pass welds. The local crack growth instability phenomena appeared simultaneously in the simulation for configuration 1, as is seen from the experimental curve. We can conclude that, by the using finite element simulation, it is possible to analyse fracture behaviour of the strength mismatched weld in detail by using global and local material properties in the elastic-plastic ductile damage model.

The numerical results also show that the local changes in tensile properties in the microstructure of an individual weld material (either in the OM or the UM) affect the load crack mouth opening curve obtained during fracture mechanical testing of the bending specimen significantly, as well as local instabilities were detected by the simulation. We conclude that, due to the random distribution of locally unequal strength areas, we will always obtain a partially different load curve; nevertheless we can describe the fracture behaviour with very good accuracy globally with the damage model.

The following conclusions appeared in this study:

The mechanical properties inside a multi pass weld region and HAZ are not constant, and this inhomogeneity should be included in FE simulation. An FE-modelling approach, where different properties inside a weld are distributed randomly, and where the inhomogeneity of the HAZ is included, shows sufficient correlation with the experimental results;Similar stiffness responses, reaction force versus CMOD between experimental and simulation were observed. Small changes in crack paths appeared due to the idealised weld geometries;FE-modelling with randomly distributed material properties should be considered further, together with simulation of similar welded structures, especially those weld connections that present a “weak-spot” for the entire structure.

## Figures and Tables

**Figure 1 materials-14-05896-f001:**
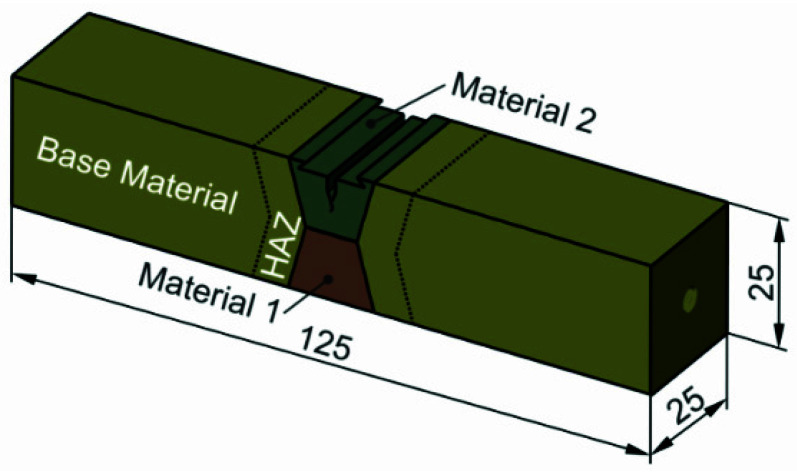
Schematic position of a mechanical notch in a bi-material welded SENB specimen.

**Figure 2 materials-14-05896-f002:**
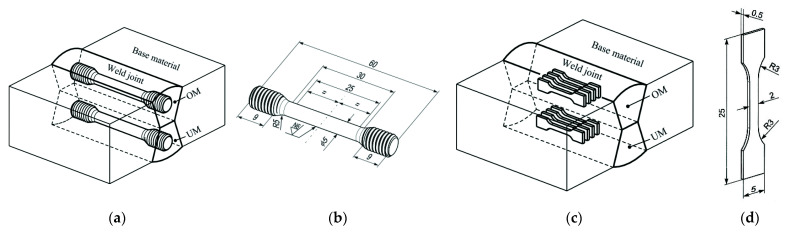
Positions of tensile specimens in weld joint and geometry of MTSs: (**a**) orientation and position of round tensile specimen in weld metal; (**b**) round specimen geometry; (**c**) orientation and position of set of mini tensile specimens in weld metal; (**d**) mini tensile specimen geometry.

**Figure 3 materials-14-05896-f003:**
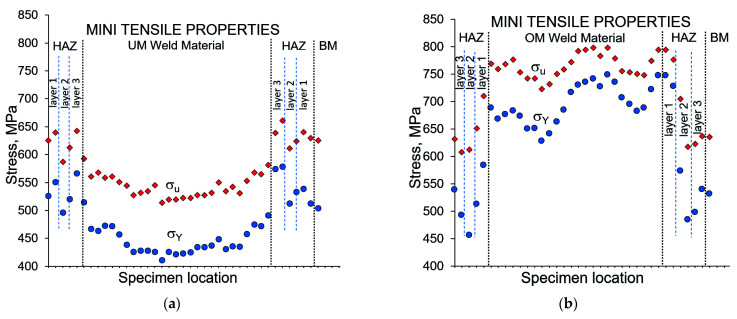
Results of tensile testing of both welded metals from mini tensile specimens: (**a**) under-match weld joint; (**b**) over-match weld joint.

**Figure 4 materials-14-05896-f004:**
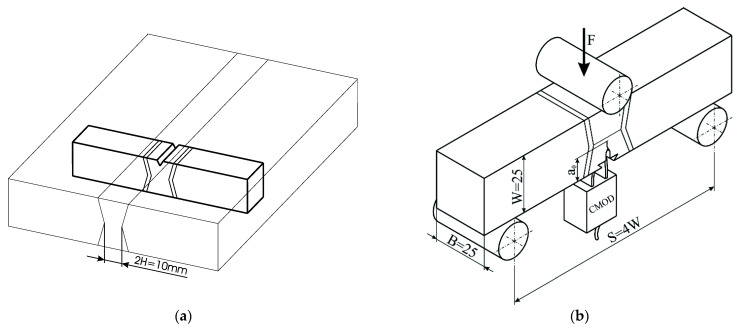
Schematic view of specimen orientation in a welded plate and testing manner: (**a**) specimen notch-crack orientation; (**b**) specimen for three point bending fracture toughness testing.

**Figure 5 materials-14-05896-f005:**
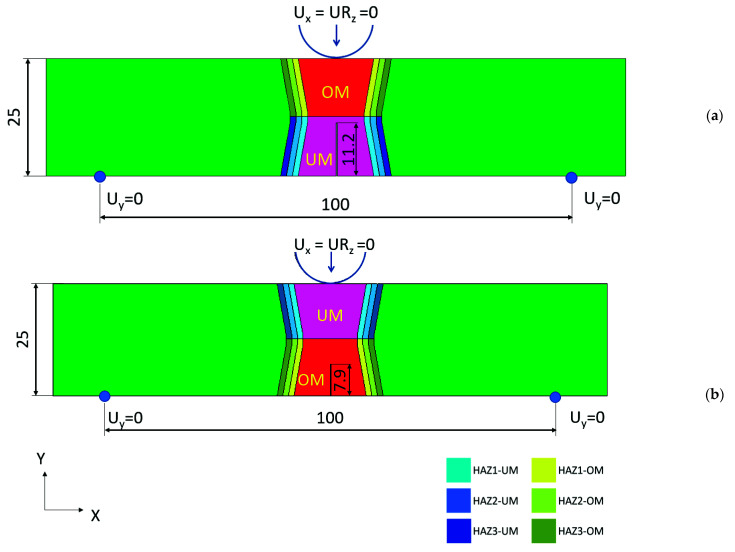
Schematic view on model for FEM with partition for each material enrolled in analysis: (**a**) UM/OM configuration 1 (initial crack in UM); (**b**) OM/UM configuration 2 (initial crack in OM).

**Figure 6 materials-14-05896-f006:**
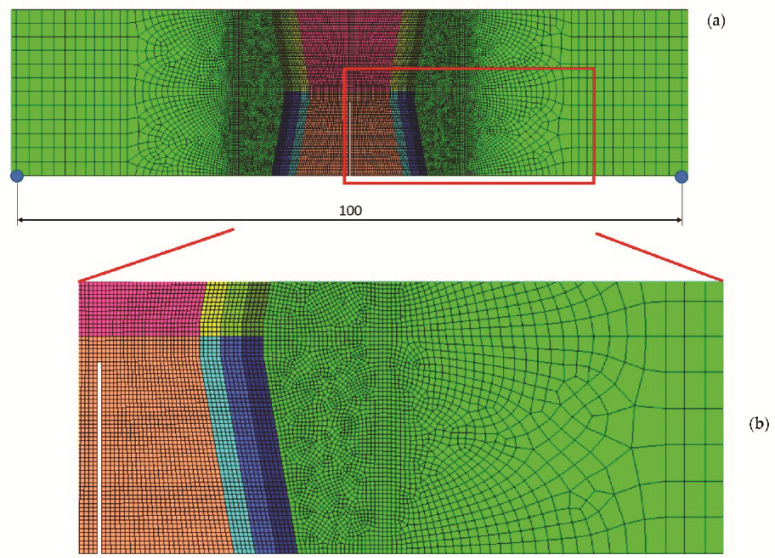
Model for FEM analysis with crack tip in the middle of the weld metal: (**a**) mesh of the entire model; (**b**) detailed mesh in the relevant area.

**Figure 7 materials-14-05896-f007:**
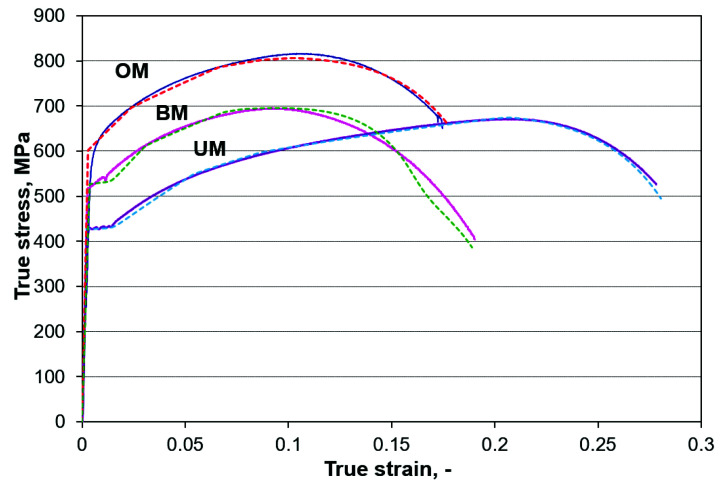
Average mechanical tensile properties obtained by standard tensile testing of round tensile specimens, as is shown in Figure 2a.

**Figure 8 materials-14-05896-f008:**
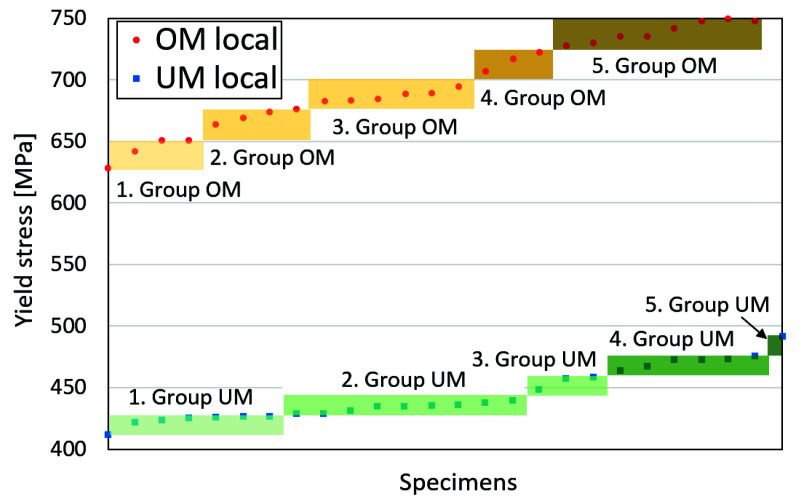
Distribution of yield stress results on 5 equidistant levels for each weld metal.

**Figure 9 materials-14-05896-f009:**
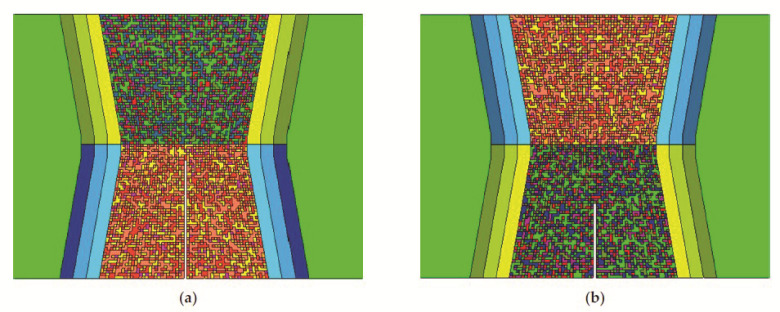
Random distribution of 5 group of mechanical properties in mesh elements for both combinations of specimens: (**a**) UM/OM configuration 1 (initial crack in UM); (**b**) OM/UM configuration 2 (initial crack in OM).

**Figure 10 materials-14-05896-f010:**
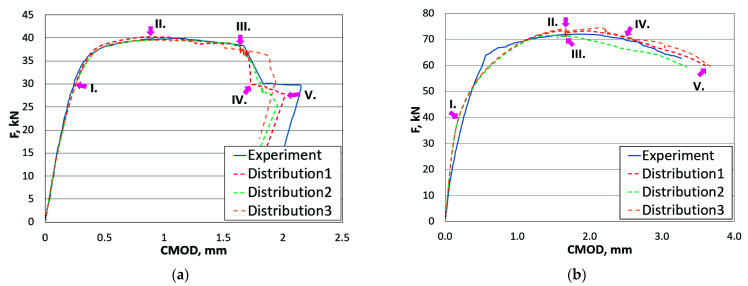
Loading curves for both specimens obtained by experimental measurements and numerical simulations: (**a**) UM/OM configuration 1 (initial crack in UM); (**b**) OM/UM configuration 2 (initial crack in OM).

**Figure 11 materials-14-05896-f011:**
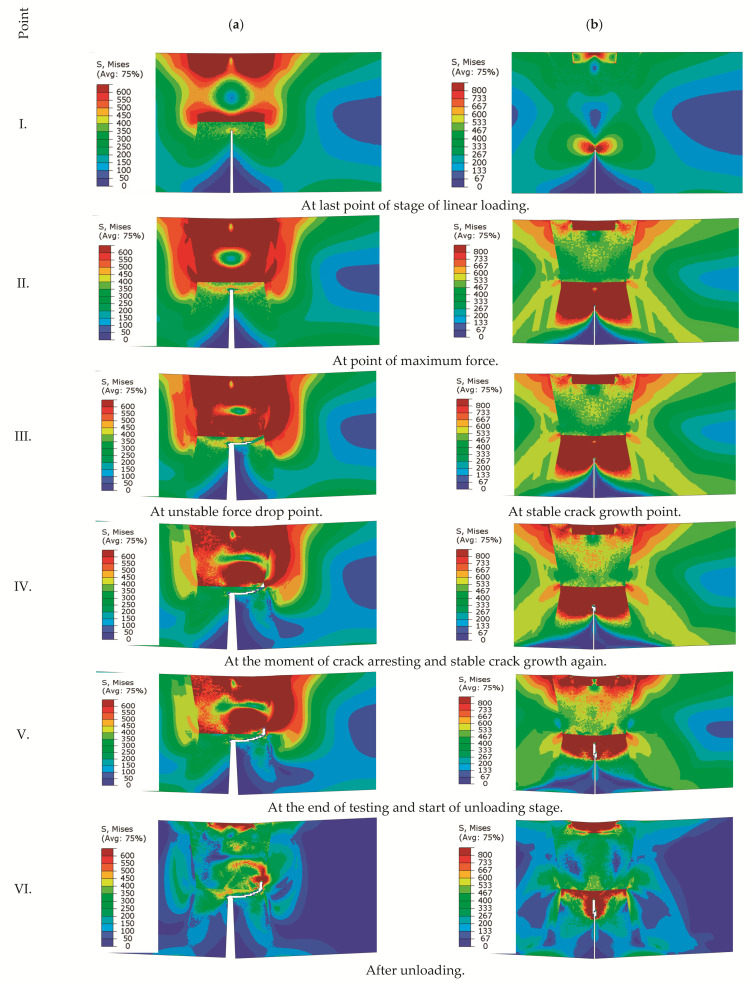
Crack path obtained by FEM simulation of both specimens in characteristic points of loading: (**a**) UM/OM configuration 1 (initial crack in UM); (**b**) OM/UM configuration 2 (initial crack in OM).

**Figure 12 materials-14-05896-f012:**
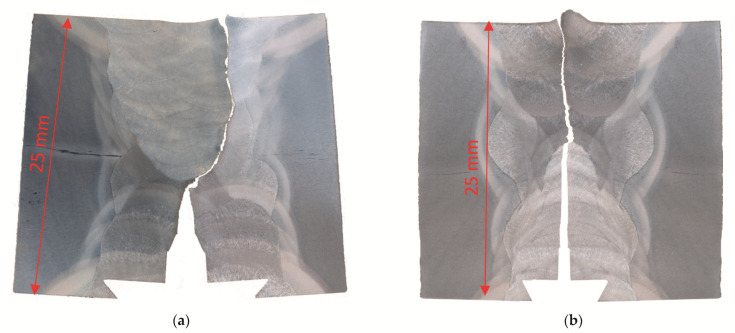
Crack path after fracture toughness testing of both specimens, etched by 4% Nital: (**a**) UM/OM configuration 1 (initial crack in UM); (**b**) OM/UM configuration 2 (initial crack in OM).

**Table 1 materials-14-05896-t001:** Tensile mechanical properties.

Configuration	Crack Growth Direction	Material 1	Material 2
I	UM → OM	OM (FILTUB 75)	UM (VAC 60)
II	OM → UM	UM (VAC 60)	OM (FILTUB 75)

**Table 2 materials-14-05896-t002:** Tensile mechanical properties.

Material	Lebel	R_p02_ [MPa]	R_m_ [MPa]	M	Charpy, Kv
Base material	NIOMOL 490	510	650	-	>60 J at −50 °C
Over matched	FILTUB 75	700	780	1.37	>40 J at −50 °C
Under matched	VAC 60	437	556	0.86	>80 J at −50 °C

**Table 3 materials-14-05896-t003:** Actual chemical composition (in weight %).

Material	C	Si	Mn	P	S	Cr	Mo	Ni
Base material	0.123	0.33	0.56	0.003	0.002	0.57	0.34	0.13
Over matched	0.040	0.16	0.95	0.011	0.021	0.49	0.42	2.06
Under matched	0.096	0.58	1.24	0.013	0.16	0.07	0.02	0.03

**Table 4 materials-14-05896-t004:** Configuration of both models for FEM analysis.

Sample Configuration	Number of Elements	Number of Nodes
1 (initial crack in UM)	18,350	18,525
2 (initial crack in OM)	18,364	18,525

**Table 5 materials-14-05896-t005:** Material parameters for reference BM, OM and UM.

Material Model	E [GPa]	ν [-]	R_p02_ [MPa]	$\varepsilon_{0}^{pl}\;[-]$	$u_{f}^{pl}\;[\mathrm{mm}]$
Base material	210.0	0.3	530	0.08	0.5
Over matched	210.0	0.3	605	0.08	0.3
Under matched	210.0	0.3	430	0.08	0.3

**Table 6 materials-14-05896-t006:** Groups of mechanical properties used in FEM analysis.

	Group	R_p02_ [MPa]	R_m_ [MPa]	Share
OM weld materials region	1	640.4	730.1	16%
	2	664.6	745.2	16%
	3	688.8	760.3	24%
	4	712.9	775.4	12%
	5	737.1	790.5	32%
UM weld materials region	1	419.5	520.2	27%
	2	435.4	533.7	35%
	3	451.4	547.3	12%
	4	467.3	560.9	23%
	5	483.2	574.4	4%
HAZ at OM side	HAZ1	578	707	
	HAZ2	471	614	
	HAZ3	539	634	
HAZ at UM side	HAZ1	573	647	
	HAZ2	504	599	
	HAZ3	545	640	

## Data Availability

Not applicable.

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
