# Peer review of "The Numerical Modelling Approach with a Random Distribution of Mechanical Properties for a Mismatched Weld"

_materials, 2021, doi:10.3390/ma14195896_

Round 1
Reviewer 1 Report
The manuscript shows an interesting topic about mechanical properties for a mismatched weld. As such it can be published in Materials considering the following comments.
- Why the NIOMOL was used as base metal?
- Do you have any structural specific detail, such as abnormal grain growth?
- Have the authors checked the reproducibility of the results?
- It is difficult to read the text inside the figures. The authors should modify.
- In Figure 6a it is impossible to see the red square. The authors should modify.
- Could you improve the discussion?
- Likewise, the authors can check the English, for minor details.
Author Response
Dear reviewer thank you very much for your comments and suggestions. We have followed your comments and provide follows answers:
- Why the NIOMOL?
Answer to the reviewer: We have put additional explanations in the text as follows:
Put in the paper:
Material NIOMOL 490K is high strength low alloy fine grain steel, with retard to coarse grain growing in the heat-affected zone (HAZ). Therefore, NIOMOL 490K has very good weldability and it is possible to weld without preheating with low strength consumable materials, e.g. VAC60.
- Do you have any structural specific detail, such as abnormal grain growth?
Answer to the reviewer: As we describe base metal retards coarse grain growth in HAZ. Both consumables have low carbon content and therefore microstructure hardness will not exceed critical values. However, in our paper, we are considering crack growth from undermatch to overmatch weld metal and vice versa under sustain loading. The initial crack tip is located in the middle of the weld joint and the grain’s size is smaller in both weld metals than HAZ.
- Have the authors checked the reproducibility of results?
Answer to the reviewer: It was done broad experimental investigation including fracture mechanical testing. Numerical reproducibility is described in the paper. Actually, it is the main point that we got similar results by numerical simulation if we are randomly distributed 5 mechanical properties values with the same friction in the total volume of weld metal.
- It is difficult to read the text inside the figures!
Figures 2, 3, 5, 8 and 10 are corrected.
- In Figure 6a, it is impossible to see the read squire.
We have made more evident read squire.
- Could you improve the discussion?
Answer to the reviewer: We have added some facts which are coming from the results!
We have put in the paper: The numerically obtained results, which were compared with the experimental fracture behaviour of three-point SENB standard specimens by ASTM E-1820, showed a good agreement on the load vs. crack mouth opening displacement curves, as well as a comparable match between the numerically simulated and metallographic measurement deviation of the crack paths. The simulation results show that the fracture behaviour of a three-point SENB specimen with a crack in the middle of a globally heterogeneous weld with good agreement with the experimental results can be described on the basis of tensile stress-strain curves obtained from standard, and scaled with Mini-Tensile Specimens, for each material microstructure.
- Likewise, the authors can check the English, for minor details.
Answer to the reviewer: Paper was corrected by native English speaks reviewer.
Reviewer 2 Report
Dear authors,
The paper entitled “The numerical modelling approach with a random distribution of mechanical properties for a mismatched weld” presents valuable results in the field of welding process. After review process there are some items unclear, namely:
Materials and experimental is not clear
Fig. 1. Which is material 1 and material 2? I supposed they represent configuration 1 and 2?
Nothing about welding process.
Two crack configurations?
Please make reference to table 2, table 4 in the text, also for figure 4. Use the same notation for figure (Figure 7 not Fig. 7 – line 176)
How were obtained Mini Tensile Specimens (MTS)
Figure 2. Positions of tensile specimens in weld joint and geometry of MTS? This figure is unclear for me. From this figure I understand that you insert the specimens into weld joint?
Author Response
Dear reviewer thank you very much for your comments and suggestions. We have followed your comments and provide follows answers:
1. Materials and experiments are not clear!
Answer: We have put additional explanations in the text as follows:
Put in the paper:
Material NIOMOL 490K is high strength low alloy fine grain steel, with retard to coarse grain growing in the heat-affected zone (HAZ). Therefore, NIOMOL 490K has very good weldability and it is possible to weld without preheating with low strength consumable materials, e.g. VAC60. The mechanical properties and chemical compositions of the BM and the OM and UM weld metals are given in Tabs. 1 and 2.
- Nothing about the welding process!
Answer: We have put additional explanations in the text as follows:
Put in the paper:
A Flux Cord Arc Welding (FCAW) procedure was applied, and two different tubular wires were selected for the welding in order to produce welded joints in over-and under-matched (OM and UM) configurations. The heat input of each weld pass was between 15-18 kJ/cm, corresponding to the cooling time between 800°-500°C Dt8/5=8-12 s.
Such weld metal configurations are common for repairing welding. When flows or pores are detected in the overmatch weld metal. The owner of the structure after reduction service tensile stress, remove materials with flaws or pores and performing weld with undermatch material without preheating.
Experiments are performed at room temperature for mini tensile test and three-point bending specimens at servo-hydraulic testing machine INSTRON. A single specimen method was used for the crack tip opening displacement (CTOD) testing according to Standard BS 7448 [6]. The CTOD tests were performed at room temperature (+24°C) under displacement control at a loading rate of 1 mm/min. The load (F), the load point displacement, the crack mouth opening displacement (CMOD). Figure 10. a) shows the experimentally obtained F vs. CMOD load curve where, after a certain amount of stable crack propagation in the OM metal and after achieving the maximum load, a step of unstable crack propagation is exhibited, followed by stable crack growth in the UM metal.
- Fig. 1 which is material 1 and 2?
Answer: We have put an additional table with explanations in Fig. 1
Bellow specimen Fig.1:
|
configuration |
Crack growth direction |
Material 1 |
Material 2 |
|
I. |
UM à OM |
OM (FILTUB 75) |
UM (VAC 60) |
|
II. |
OM à UM |
UM (VAC 60) |
OM (FILTUB 75) |
- Two crack configurations:
Answer: Yes, two configurations with respect to material where the crack tip was positioned. We have put additional explanations in the text as follows:
Put in the paper: In the paper are considered two specimens with different configurations with respect to material where the crack tip was positioned as is shown in Fig. 1.
- Please make reference to table 2
Answer:
Thank you. We have referred to all mentioned Tables in the text of the manuscript.
- How were obtained Mini tensile specimens?
Answer:
Cut positions of tensile specimens in weld joint and geometry of MTS are shown in Figure 2. MTS specimens are made by wire spark eroding techniques.
Round 2
Reviewer 2 Report
Dear Authors,
Thank you for your clear explanations. The paper was considerable improved and I agree with its publication.
Best regards!